# An updated census of the maize *TIFY* family

**Pingdong Sun**[1,2], **Yannan Shi**[1], **Aga Guido Okwana Valerio**[1], **Eli James Borrego**[3], **Qingyun Luo**[4], **Jia Qin**[1], **Kang Liu**[1], **Yuanxin Yan**[1,5]*

1 State Key Laboratory for Crop Genetics and Germplasm Enhancement, Nanjing Agricultural University, Nanjing, China, 2 Crop Breeding & Cultivation Research Institution, Shanghai Academy of Agricultural Sciences, Shanghai, China, 3 Thomas H. Gosnell School of Life Sciences, Rochester Institute of Technology, Rochester, NY, United States of America, 4 College of Horticulture, Nanjing Agricultural University, Nanjing, China, 5 Jiangsu Collaborative Innovation Center for Modern Crop Production, Nanjing Agricultural University, Nanjing, China

* yuanxin.yan@njau.edu.cn

**Data Availability Statement:** All relevant data are within the paper and its Supporting Information files.

**Funding:** This research was supported by the National Natural Science Foundation of China (Grant No. 31571580), the Fundamental Research

## Abstract

The *TIFY* gene family is a plant-specific gene family encoding a group of proteins characterized by its namesake, the conservative TIFY domain and members can be organized into four subfamilies: ZML, TIFY, PPD and JAZ (Jasmonate ZIM-domain protein) by presence of additional conserved domains. The *TIFY* gene family is intensively explored in several model and agriculturally important crop species and here, yet the composition of the *TIFY* family of maize has remained unresolved. This study increases the number of maize TIFY family members known by 40%, bringing the total to 47 including 38 *JAZ*, 5 *TIFY*, and 4 *ZML* genes. The majority of the newly identified genes were belonging to the *JAZ* subfamily, six of which had aberrant TIFY domains, suggesting loss JAZ-JAZ or JAZ-NINJA interactions. Six *JAZ* genes were found to have truncated Jas domain or an altered degron motif, suggesting resistance to classical JAZ degradation. In addition, seven membranes were found to have an LxLxL-type EAR motif which allows them to recruit TPL/TPP co-repressors directly without association to NINJA. Expression analysis revealed that *ZmJAZ14* was specifically expressed in the seeds and *ZmJAZ19* and *22* in the anthers, while the majority of other *ZmJAZs* were generally highly expressed across diverse tissue types. Additionally, *ZmJAZ* genes were highly responsive to wounding and JA treatment. This study provides a comprehensive update of the maize *TIFY/JAZ* gene family paving the way for functional, physiological, and ecological analysis.

## Introduction

Jasmonates (JAs) are plant oxylipin hormones involved in the regulation of diverse physiological processes in plants, including reproductive development, abiotic stress response, and defense against insect and microbes [1–3]. In plant cells, jasmonates are synthesized from linolenic acid via the octadecanoid pathway [4–6], through the activity of at least eight enzymes (lipase, lipoxygenase, allene oxide synthase and cyclase, 12-OPDA (12-oxophytodienoic acid) reductase, acyl-CoA oxidase, a multifunctional protein, and 3-ketoacyl-CoA thiolase) [7–9]. JA perception occurs through the interaction of the biologically active ligand, JA-Ile, with

Funds for the Central Universities (KYTZ201402 and KYRC201404), the Outstanding Scientific Innovation Team Program for Jiangsu Universities (2015), the Agricultural Applied Technology Development Program of Shanghai City (Z20180103), the Technology-Promoting Program Funded by Science and Technology Commission of Shanghai Municipality (17391900100) and the Youth Talent Career-Development Plan Funded by the commission of Agriculture and Rural Affairs of Shanghai Municipality (20180102).

**Competing interests:** The authors have declared that no competing interests exist.

SCF$^{COI1}$ which results in ubiquitination of the JAZMONATE ZIM-Domain (JAZ) transcriptional repressors that are then targeted for degradation by the 26S proteasome proteolytic pathway [10,11]. The result is derepression of bHLH transcription factors, such as MYC2, allowing activation of JA responsive gene induction [10–12].

JAZs belong to the larger TIFY superfamily [13], previously known as Zinc-finger protein expressed in the inflorescence meristem (ZIM) [14]. The TIFY family members contain the TIFY motif and are grouped into four subfamilies: ZML (ZIM-like), TIFY, PPD, and JAZ based on their domain structure [13,15]. The members of the ZML subfamily contain a TIFY, C2C2-GATA zinc-finger, and CCT domain [16]. Proteins unified with only the TIFY motif belong to the TIFY subfamily [13]. PPD proteins possess three domains: an N-terminal PPD domain, a TIFY domain, and a Jas domain located near the N-terminus [15]. The JAZ subfamily members have two conserved domains: the TIFY domain at the N-terminal with the core sequence TIF [F/Y] XG, and a Jas domain at the C-terminal with a unique sequence SLX2FX2KRX2RX5PY [12,13,17]. Unlike the variable TIFY domain, the sequence of the Jas domain is remarkably conserved among all JAZ subfamily members across different plant species. Many JAZ isoforms are characterized as transcriptional repressors and are commonly associated with co-repressors such as TOPLESS (TPL)/TPL-related proteins (TPPs) that interact with the adaptor protein, NOVEL INTERACTOR OF JAZ (NINJA) [18]. In the absence of JA, the TIFY domain interacts with the C-terminal of NINJA while the Jas domain binds and represses bHLH transcription factors [12,18–20].

In recent years, JAZ proteins have been intensively investigated, primarily for their roles in numerous aspects of plant development and defense responses against biotic and abiotic stresses. Gain-of-function mutations in AtJAZ2 prevent coronatine-mediated stomatal reopening and are highly resistant to *Pseudomonas syringae* [21]. AtJAZ1, AtJAZ3, and AtJAZ4 interact with APETALA2 transcription factors to repress the transcription of FLOW-ERING LOCUS *T* (FT) [22]. AtJAZ7 negatively regulates dark-induced leaf senescence [23]. Additionally, AtJAZ7, along with AtJAZ8, play a role during defense to fungal infection and insect herbivory [24,25]. AtJAZ1 and AtJAZ10 are among the best understood JAZs owed to their repression of the well-explored JA-responsive transcription factor, AtMYC2 [26,27] and AtJAZ13 has also been found to physically interact with JA-responsive transcriptional factor AtMYC2 [28]. A recent study discovered that JAZ proteins promote growth and reproduction by preventing unnecessary plant immune responses [29].

Most JAZ genes explored thus far are wound- and herbivory-inducible [30]. In rice, overexpression of *JAZ* genes with a mutated Jas domain such as *mOsJAZ3*, *mOsJAZ6*, *mOsJAZ7*, and *mOsJAZ11* affect spikelet development and have wide-spread pleiotropic effects [31]. The overexpression of *OsJAZ9* increases tolerance to salt and drought [32]. In tomato, several *JAZ* genes are inducible by pharmacological application of JA and abscisic acid (ABA) and *SlJAZ*3, *SlJAZ*7, and *SlJAZ*10 in particular are induced in leaves following salt treatment [33]. Together, these studies provide convincing evidence highlighting the importance of JAZ proteins in plant development, growth, and defense.

In recent years, the genomes of many plant species have been surveyed to catalogue their *TIFY/JAZ* genes. In *Arabidopsis*, 19 members constitute the TIFY family which includes two *ZML*, two *PPD*, two *TIFY* and 13 *JAZ* genes [15,28,30,34]. Comparative analysis of other plant species found variability in their *TIFY* genes content with *Arabidopsis* [13,28], tomato [33], Asian cotton [35], *Brachypodium distachyon* [36], Chinese pear [37], grape [38], *Brassica napus* [39], rice [32], maize [15,40–42], and wheat [43] containing 19, 19, 21, 21, 22, 19, 36, 20, 30, and 47 TIFY members, respectively. In these species, JAZs account for about 66% of TIFY family [15]. Interestingly, in the monocotyledonous species no PPD proteins have been identified so far [15]. In maize, the literature has yet to reach an agreement over the accurate number

of total *TIFY* genes where as little as 27 to as high as 48 were reported [15,40–42]. In 2016, the maize reference genome was updated using single-molecule sequencing technology to Zm-B73-REFERENCE-GRAMENE-4.0 (also known as "B73 RefGen_v4" or "AGPv4") which is substantially different from the previous AGPv3. In this study, we utilized version 4 of the maize reference genome to update the list of *TIFY* genes and classified them into subfamilies based on the presence of their respective conservative domains. To provide insights into the functions of different family members, the expression of all the *ZmTIFY* genes were assessed in various tissues and organs at different developmental stages and in response to wounding and JA chemical treatment. In addition, the promoters of *ZmTIFY* genes were analyzed for predicted *cis*-elements that may explain potential conditional-dependent gene induction.

## Materials and methods

### Plant material

The maize inbred B73 was used as the plant material for this study. The seeds were sowed in plastic boxes containing a soil mix of vermiculite: organic substrate: loam (1:1:1 v/v/v). The seedlings were grown in a greenhouse at 25–35 C with relative humidity maintained at 60%-85% and illuminated by natural sunlight. The experiments were carried out in the seasons of Spring or Autumn when the average photoperiods were approximate 12 h-day/12 h-night.

### Mechanical wounding and JA treatment

The mechanical wounding treatment was conducted as described by [44]. The second leaf of a V3 stage plants was squeezed with pliers twice on each side of the midrib about 1cm apart without damaging to the midrib. The undamaged midsection flanked by the two wound sites was collected at 0, 1, 3, 6 h post-wounding and frozen immediately in liquid nitrogen and stored at -80˚C for downstream analysis.

Seedlings at the V3 stage were sprayed with 100μM of JA solution or water as control until both sides of the leaves were completely wet and collected at 0, 6, 12, 24, and 48 h after chemical treatment, frozen immediately in liquid nitrogen, and stored at -80˚C until further analysis. Three biological replicates were collected per time-point for each treatment-group.

### Gene expression analysis

Total RNA was extracted using Trizol according to the manufacturer's instructions and its integrity was tested on a 1% agarose gel by visualizing defined 16S and 18S rRNA bands. Genomic DNA was removed according to Goldenstar™ RT6 cDNA synthesis kit (Sangon Biotech Co. Ltd at Shanghai). For reverse transcription, 2 μg of total RNA was used to generate cDNA through the Goldenstar™ RT6 cDNA synthesis kit according to the manufacturer's instructions. The cDNA synthesis reactions consisted of 2μl (~2μg) of RNA template, 4μl of Goldenstar™ RT6 cDNA synthesis mix, and 14μl of RNase-free water followed with incubation at 50˚C for 30 minutes and then at 85˚C for 10 minutes.

Expression analysis was conducted with semi-quantitative real-time PCR using primers designed to selectively amplify distinct JAZs (S3 Table) and *EIF4A* gene was used as the housekeeping gene control for equal loading of cDNA. The reaction consisted of 12.5μl of 2xTaq PCR master mix (Sangon Biotech Co. Ltd at Shanghai), 1 μl forward primer (10μM), 1μl reverse primer (10μM), 1 μl (100 ng) of cDNA and ddH2O to a final volume of 25μl. Thermal cycling conditions were: 94˚C for 4 mins; 94˚C for 30 s, 57–58˚C for 30 s, and 72˚C for 30 s, a final incubation at 72˚C for 10 min, and depending on reaction, 28–30 cycles were performed. The PCR products were separated and visualized by gel electrophoresis on a 2% agarose gel.

## Identification of the maize *TIFY* gene family and domain analysis

To identify the members of the *TIFY* family in maize, BLASTP searches were performed on the maize genome database (B73 RefGen_v4, https://maizegdb.org/) using the amino acid (AA) sequences of TIFY and Jas domains from TIFY proteins from *Arabidopsis* and rice as the search queries. Maize *TIFY* candidate genes were selected based on the criteria of 50% or greater AA identity and an e-value of 1e-4 or less. To determine the presence of the canonical TIFY subfamily domains, the predicted AA sequences of the *ZmTIFY* genes were submitted to the Pfam database (http://pfam.xfam.org/). For the analysis of the presence of an EAR (ERF-associated amphiphilic repression) motif, candidate proteins were manually compared to the previously reported 158 LxLxL-types of EAR motifs [45].

## Tissue-specific expression profiling

RNA-Seq data for tissue-specific expression in 79 tissues [46] were obtained from maizegdb. org. The expression heatmap for tissue-specific expression was created by the software HemI 1.0 [47] using $\log_2$ value of FPKM (fragment per kilobase per million mapped reads) of *ZmTIFY* genes.

## Phylogenetic analysis of *TIFY* genes

A multiple protein sequence alignment was performed for the TIFY family members of *Arabidopsis*, maize, and sorghum using the online software MUSCLE (www.ebi.ac.uk/Tools/msa/muscle/). The phylogenetic tree for all identified TIFY family genes in this study and for all known JAZ genes in *Arabidopsis* and sorghum were generated with the MEGA 7.0 software using the maximum likelihood method and robustness tested by bootstrapping for 1000 times. The tree was displayed using the online software Evolview v3 [48].

## *cis*-element identification in promoters of *TIFY* genes

To analyze the putative *cis*-acting elements of the promoters of the *ZmJAZ* genes, 1.5 kb of nucleotide sequence upstream of the start codon for each *ZmJAZ* gene was scanned in the PlantCARE database (http://bioinformatics.psb.ugent.be/webtools/plantcare/html/).

## Results

### The maize genome houses 47 bona fide TIFY family members including 16 newly identified members

To identify all *TIFY* genes in the maize genome, the B73 RefGen_v4 genome was surveyed by BLASTP for similar sequences to the AA sequences of the TIFY and Jas domains from the *Arabidopsis* and rice TIFY proteins. This analysis revealed 47 distinct gene models whose predicted proteins contain a TIFY or Jas domain (Tables 1 and S1). Among these, four were predicted to belong to the ZML subfamily and contained a TIFY, CCT, and GATA zinc finger domain, but no Jas domain. Five of 47 were predicted to belong to the TIFY subfamily which contains solely the TIFY domain (Tables 1 and S1). No PPD proteins were identified. The remaining 38 TIFY proteins were characterized as JAZ proteins, six of which had no TIFY domain at the N-terminus, but all had a Jas domain at the C-terminus (Tables 1 and S1). In total, 38 JAZ, 4 ZML, 5 TIFY-subfamily, and no PPD genes were identified in the maize genome B73 RefGen_v4. Among the 47 *TIFY* genes, nearly 40% have never been identified in previous analyses of the maize TIFY family. These 16 genes include 13 *ZmJAZs*, two *ZmTIFYs* and one *ZmZMLs* (Tables 1 and S1).

**Table 1.  List of *TIFY* family genes in maize.**

| Locus ID in (V4) | Locus ID in (V3) | Chromosomal Location (V4) | Gene Name [a] | Gene Name [b] | Transcript Length (bp, V4) | Protein Length (aa, V4) | TIFY motif [c] | Jas Domain |
|---|---|---|---|---|---|---|---|---|
| Zm00001d027899 | GRMZM2G343157 | 1:17141137 | zim26 | *ZmJAZ1* | 495 | 164 | TILYGG | Yes |
| Zm00001d027901 | GRMZM2G445634 | 1:17156322 | zim16 | *ZmJAZ2* | 546 | 181 | TIFYGG | Yes |
| Zm00001d029448 | GRMZM2G117513 | 1:71161670 | zim24 | *ZmJAZ3* | 687 | 228 | TIFYGG | Yes |
| Zm00001d033048 | GRMZM2G024680 | 1:248467942 | zim21 | *ZmJAZ4* | 651 | 216 | TIFYQG | Yes |
| Zm00001d033050 | GRMZM2G145412 | 1:248529926 | zim18 | *ZmJAZ5* | 549 | 182 | TIVYGG | Yes |
| Zm00001d033049 | GRMZM2G145458 | 1:248522876 | zim3 | *ZmJAZ6* | 489 | 162 | TISYGG | Yes |
| Zm00001d034536 | GRMZM2G382794 | 1:295853517 | zim19 | *ZmJAZ7* | 357 | 176 | TIFYGG | Yes |
| Zm00001d002029 | GRMZM2G086920 | 2:4666311 | zim32 | *ZmJAZ8* | 558 | 216 | TIFYGG | Yes |
| Zm00001d003903 | GRMZM2G145407 | 2:66485018 | zim33 | *ZmJAZ9* | 543 | 180 | TVFYGG | Yes |
| Zm00001d004277 | GRMZM2G171830 | 2:99601657 | zim8 | *ZmJAZ10* | 297 | 134 | TIFYDG | Yes |
| Zm00001d005813 | GRMZM2G005954 | 2:189505960 | zim13 | *ZmJAZ11* | 531 | 227 | TIFYGG | Yes |
| Zm00001d006860 | GRMZM2G101769 | 2:218018545 | zim12 | *ZmJAZ12* | 744 | 237 | TIFYGG | Yes |
| Zm00001d050365 | GRMZM2G151519 | 4:83772143 | zim35 | *ZmJAZ13* | 1281 | 426 | TIFYNG | Yes |
| Zm00001d014249 | GRMZM2G064775 | 5:38005178 | zim29 | *ZmJAZ14* | 657 | 218 | TIFYQG | Yes |
| Zm00001d014253 | GRMZM2G173596 | 5:38196209 | zim10 | *ZmJAZ15* | 483 | 160 | IIVYGG | Yes |
| Zm00001d035382 | GRMZM2G338829 | 6:23840275 | zim9 | *ZmJAZ16* | 507 | 110 | TIFYGG | Yes |
| Zm00001d020409 | GRMZM2G126507 | 7:112014245 | zim1 | *ZmJAZ17* | 1215 | 404 | TIFYAG | Yes |
| Zm00001d020614 | GRMZM2G116614 | 7:125133740 | zim28 | *ZmJAZ18* | 657 | 218 | TIFYGG | Yes |
| Zm00001d021274 | GRMZM2G066020 | 7: 147534788 | zim31 | *ZmJAZ19* | 657 | 267 | TIFYGG | Yes |
| Zm00001d022139 | GRMZM2G089736 | 7:171049645 | zim23 | *ZmJAZ20* | 702 | 233 | TIFYGG | Yes |
| Zm00001d048263 | GRMZM2G036351 | 9:153418013 | zim4 | *ZmJAZ21* | 519 | 172 | TIFYGG | Yes |
| Zm00001d048268 | GRMZM2G036288 | 9:153485703 | zim14 | *ZmJAZ22* | 552 | 183 | TIFYGG | Yes |
| Zm00001d026477 | GRMZM2G143402 | 10:146705762 | zim34 | *ZmJAZ23* | 693 | 207 | TIFYGG | Yes |
| Zm00001d009438 | GRMZM2G054689 | 8:64583138 | zim5 | ***ZmJAZ24*** | 507 | 253 | **TIFYGG** | Yes |
| Zm00001d013855 | GRMZM2G063632 | 5:22766950 | zim7 | ***ZmJAZ25*** | 669 | 155 | **LQFSMV** | Yes |
| Zm00001d005726 | GRMZM2G114681 | 2:184842614 | zim15 | ***ZmJAZ26*** | 1620 | 353 | **TIFYAG** | Yes |
| Zm00001d027900 | GRMZM5G838098 | 4:1:17147073 | zim27 | ***ZmJAZ27*** | 609 | 195 | **TIFYGG** | Yes |
| Zm00001d014250 | AC197764.4_FG003 | 5:38073928 | zim30 | ***ZmJAZ28*** | 555 | 184 | **TLSIFY** | Yes |
| Zm00001d016316 | NO | 5:156926728 | zim37 | ***ZmJAZ29*** | 474 | 157 | **NO** | Yes |
| Zm00001d019692 | NO | 7:51184119 | zim38 | ***ZmJAZ30*** | 1353 | 98 | **NO** | Yes |
| Zm00001d021924 | NO | 7:165961049 | zim39 | ***ZmJAZ31*** | 414 | 137 | **NO** | Yes |
| Zm00001d024455 | GRMZM2G442458 | 10:71687709 | zim40 | ***ZmJAZ32*** | 183 | 60 | **NO** | Yes |
| Zm00001d033972 | NO | 1:279900021 | zim41 | ***ZmJAZ33*** | 502 | 173 | **TIFYGG** | Yes |
| Zm00001d041045 | NO | 3:92630179 | zim42 | ***ZmJAZ34*** | 621 | 206 | **TIFYGG** | Yes |
| Zm00001d044708 | NO | 3:235521147 | zim43 | ***ZmJAZ35*** | 453 | 150 | **TIFYGG** | Yes |
| Zm00001d046270 | NO | 9:77365055 | zim44 | ***ZmJAZ36*** | 414 | 137 | **NO** | Yes |
| NO | GRMZM2G327263 | 3:231288810(V3) | zim17 | *ZmJAZ37* | 4815 | 1604 | TIFYGG | Yes |
| Zm00001d037082 | GRMZM2G314145 | 6:111655145 | zim25 | *ZmJAZ38* | 1000 | 135 | NO | Yes |
| Zm00001d028313 | GRMZM2G110131 | 1:30342336 | zim22 | *ZmTIFY1* | 381 | 215 | TIFYGG | NO |
| Zm00001d004173 | NO | 2:89164906 | zim45 | ***ZmTIFY2*** | 639 | 212 | **TIFYGG** | NO |
| Zm00001d051615 | GRMZM2G022514 | 4:164594515 | zim46 | ***ZmTIFY3*** | 3306 | 1101 | **TIFYGG** | NO |
| NO | GRMZM2G036349 | 9:150516983(V3) | zim6 | *ZmTIFY4* | 411 | 136 | NO | NO |
| NO | GRMZM2G122160 | 4:11372450(V3) | zim11 | *ZmTIFY5* | 1024 | 197 | TIFYGG | NO |
| Zm00001d013331 | GRMZM2G065896 | 5:8803187 | zim2 | *ZmZML1* | 837 | 278 | TLVYQG | NO |
| Zm00001d014656 | GRMZM2G058479 | 5:57723133 | zim36 | *ZmZML2* | 882 | 357 | TLSFQG | NO |
| Zm00001d036494 | GRMZM2G080509 | 6:90506221 | zim20 | *ZmZML3* | 1077 | 357 | TLSFQG | NO |

(*Continued*)

**Table 1.** (Continued)

| Locus ID in (V4) | Locus ID in (V3) | Chromosomal Location (V4) | Gene Name [a] | Gene Name [b] | Transcript Length (bp, V4) | Protein Length (aa, V4) | TIFY motif [c] | Jas Domain |
|---|---|---|---|---|---|---|---|---|
| Zm00001d033523 | NO | 1:265546924 | zim47 | ***ZmZML4*** | 867 | 288 | **TLVFQG** | NO |

[a] The official names designated by maizeGDB and it is applied in Grassius project [40].

[b] The gene names in bold are the newly found *TIFY* genes in this study using B73 RefGen_V4.

[c] Six *TIFY* genes have no typical TIFY domain but include Jas motif. The TIFY motif of *ZmJAZ25* was largely altered, and *ZmTIFY4* lacks TIFY domain and Jas motif.

## JAZ proteins are asymmetrically distributed between the two maize subgenomes

The 47 *TIFY* genes were found differentially distributed across the ten maize chromosomes. The four *ZML* genes were located on the chromosome 1, 5, and 6 and the five TIFY-subfamily genes were found on chromosome 1, 2, 4, and 9. The remaining 38 *JAZ* genes were found on distributed across all ten chromosomes (Fig 1; Table 1). Chromosome 1 was found to contain nine *JAZ* genes, six of which were clustered in two loci: *ZmJAZ1*, *2*, and *27* and *ZmJAZ4*, *5*, and *6*. *ZmJAZ14*, *15*, and *26* were clustered at the short arm of Chromosome 5. Maize is a paleopolyploid plant, which harbors two subgenomes (maize1 and maize2) where each constitutes a genome orthologous to the entire sorghum genome [49]. Interestingly, 32 *TIFY* genes were found in the maize1, 14 in the maize2, and only one in the region between the two subgenomes (Fig 1) compared to the 19 *SbTIFY* genes thus so far predicted in the sorghum genome [15].

## The maize *TIFY* gene family members possess considerable variability in gene size, structure, and predicted transcript variants

The maize TIFY genes ranged from 474 bp (*ZmJAZ29*) to 13091 bp (*ZmJAZ37*) (S1 Document). Gene structural analysis of the TIFY family found that approximately 20% of the

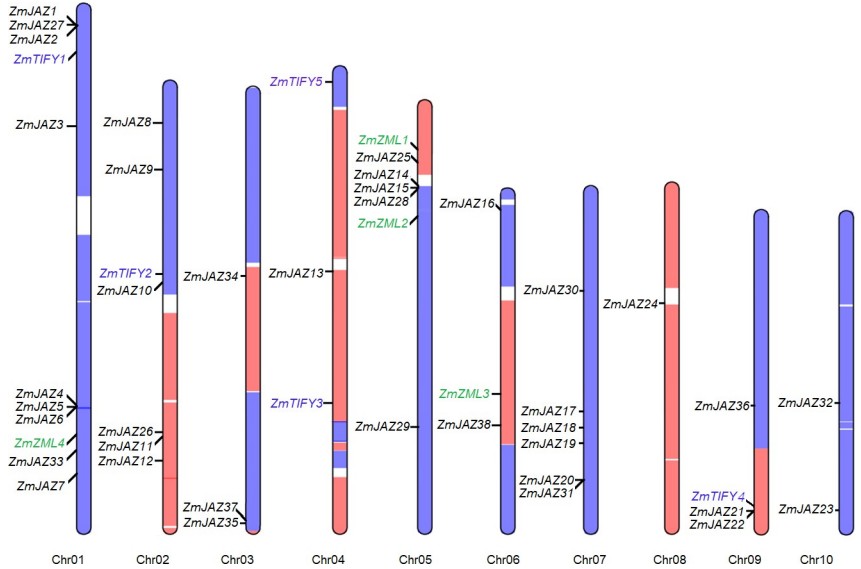

**Fig 1. Distribution of the *ZmTIFY* genes on maize chromosomes.** The blue and orange regions denote the subgenome1 (maize1) and subgenome2 (maize2) of maize genome (Schnable et al., 2011) and the newly found *TIFY* genes are highlighted in blue.

members (*ZmJAZ1*, *2*, *5*, *6*, *15*, *21*, *22*, *29*, and *34*) were comprised of a single exon and nearly 80% of TIFYs contained two to twenty one exons (Fig 2). Among the 29 *JAZs* with multiple exons, *ZmJAZ37* (v3) is a longest gene with 13091 nucleotides and 21 exons. This gene was remodeled in AGPv4 and AGPv5. The latest gene model of *ZmJAZ37* (Zm00001e020904) is only 1476-bp long containing 4 exons (S1 Document). *ZmJAZ19* had nine exons and *ZmJAZ13* and *17* each had seven exons. Eight *ZmJAZs* (*ZmJAZ3*, *8*, *11*, *12*, *18*, *20*, *24*, and *35*), three *ZmJAZs* (*ZmJAZ9*, *25*, and *33*), six *ZmJAZs* (*ZmJAZ 14*, *28*, 30, *31*, *36* and *38*), and four *ZmJAZs* (*ZmJAZ4*, *16*, *27*, and *32*) contained five, four, three, and two exons, respectively (Fig 2). The members of the TIFY subfamily contained between 3 to 10 exons and the ZML sub-family members contained either seven or eight exons (Fig 2).

The maize *TIFY* genes were predicted to have varied numbers of transcripts variants (S1 Table). Over 60% or 31 *TIFY* genes (*ZmJAZ1*, *2*, *4*, *5*, *6*, *7*, *9*, *10*, *14*, *15*, *16*, *19*, *21*, *22*, *24*, *26*, *27*, *28*, *29*, *31*, *32*, *34*, *35*, *36*, *37 and 38*, *ZmTIFY1*, *ZmTIFY2*, *ZmTIFY4*, *ZmTIFY5* and *ZmZML1*) have a single transcript (S1 Table) while the other 16 *TIFY* genes have two to eight transcripts. It is worthy to note that *ZmJAZ23* and *ZmZML2* have seven and eight predicted transcript variants, respectively (S1 Table).

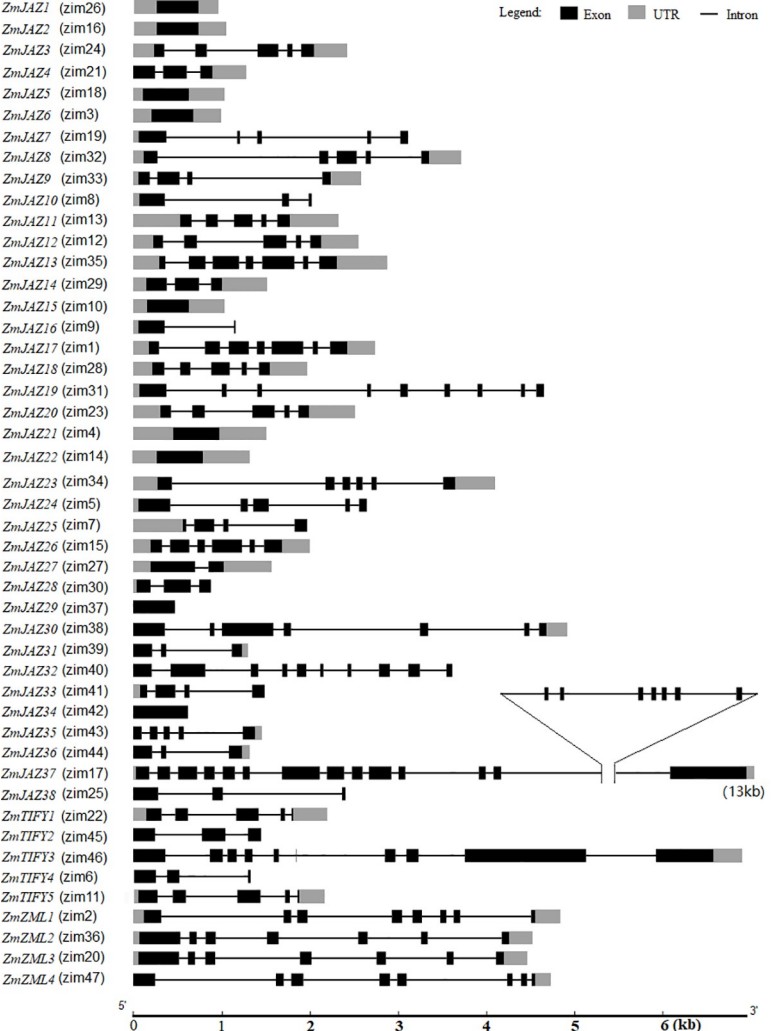

**Fig 2. The genetic structure of the maize *TIFY* genes.** Scale bar indicates gene size.

## Conserved protein domains and their features different across the maize TIFY family

All *ZmTIFY* genes are predicted to encode at least one protein which range in sizes from 60 AA to 1604 AA, however the vast majority of TIFY proteins are smaller than 300 AA (S1 Table). The candidate maize TIFY proteins were analyzed for their TIFY and Jas domain compositions along with screening for presence of EAR-motifs. The TIFY domain, also known as the ZIM domain, mediates homo- and heteromeric interactions between JAZ proteins [17,50] and it is necessary for binding to the NINJA–TPL repressor complex [18]. The C-terminal Jas domain is essential for the interaction of JAZ proteins [12] with the LRR domain of JA receptor, COI1 protein [51]. The EAR (ERF-associated amphiphilic repression) motif is a principle mechanism of plant gene regulation and facilitates recruitment of TPL for transcriptional repression [24]. The analysis revealed that all but seven TIFY proteins contained both the TIFY and the Jas domains. The TIFY motif was absent in ZmJAZ 29, 30, 31, 32, and 36, and while it was truncated in ZmJAZ25 and ZmZML4 (S1 Table; Fig 3). JAZ4, 10, and 14 had incomplete Jas domains (S1 Table) and lacked the X5PY. X5PY motif required for JAZ

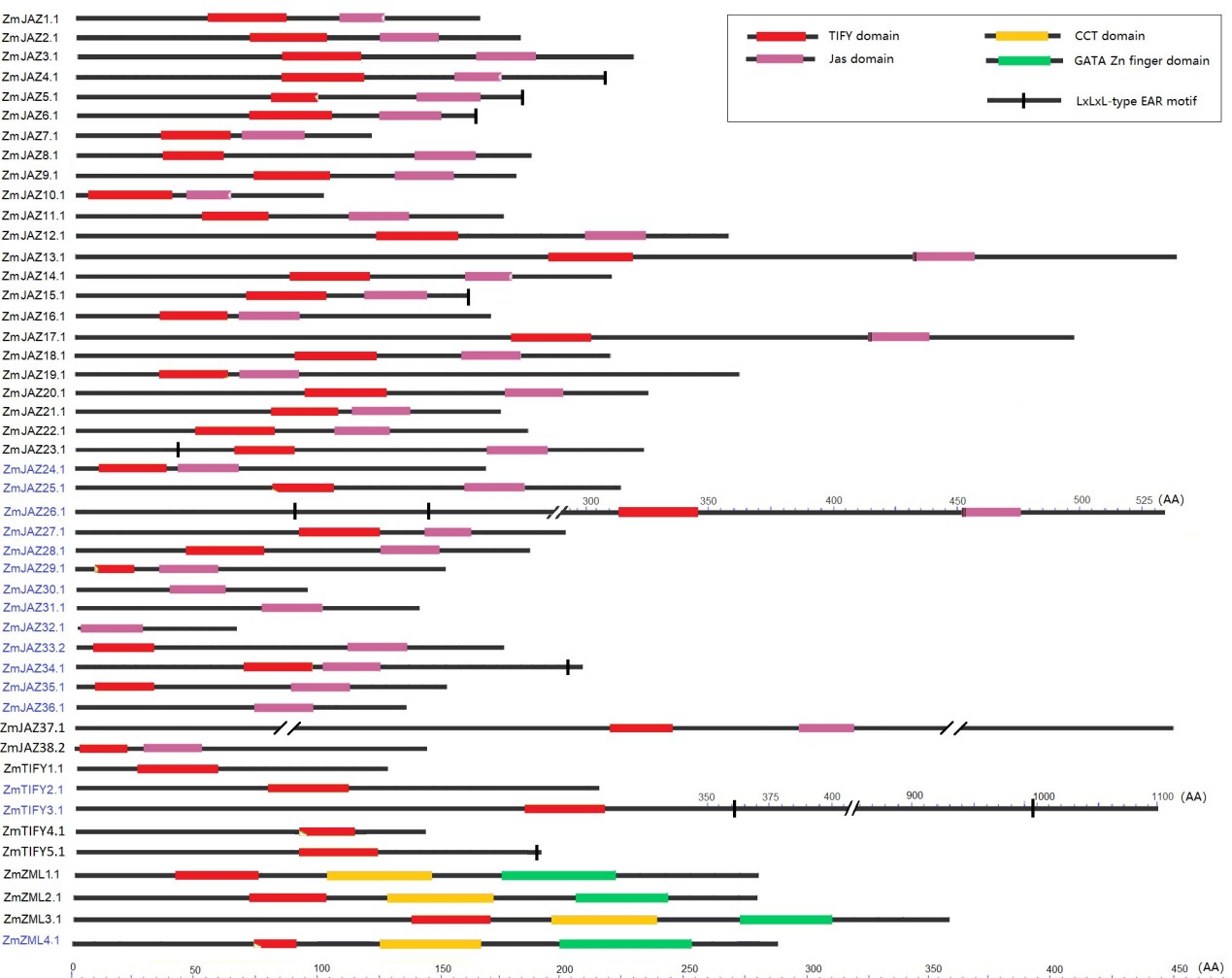

**Fig 3. Conserved domain analysis of the maize JAZ, TIFY and ZML subfamily proteins.** Each domain is represented by a colored box and black lines represented the non-conserved sequences. Scale bar represents peptide length.

degradation via 26S proteasome [12]. The Jas domains of ZmJAZ13, 17, and 26 had VPQAR in place of the normal LPIAR degron motif the sequence signal required for JAZ repressor degredation [24]. Manual sequence analysis uncovered that seven *ZmJAZs* (*ZmJAZ4*, *5*, *6*, *15*, *23*, *26* and *34*) along with *ZmTIFY3* and *5* possessed the LxLxL-type EAR motif (Fig 3).

In summary, among 38 JAZ proteins, 13 (ZmJAZ4, 10, 13, 14, 17, 25, 26, 29, 30, 31, 32, 34 and 36) had an altered or impaired TIFY or Jas domain. The five TIFY subfamily proteins (ZmTIFY1, 2, 3, 4 and 5) have showed typical TIFY domain sequences. Three of the four maize ZML proteins (ZmZML1, 2, and 3) have an intact TIFY domain, a CCT domain and a GATA zinc finger domain, however the newly identified ZmZML4 bears a truncated TIFY domain.

## The maize TIFY family members cluster into six distinct clades

To understand the evolutionary relationship of *TIFY* genes in maize, a phylogenetic tree of ZmTIFY proteins was created by the software Mega7 using the maximum likelihood method. The phylogenetic tree showed that all the TIFY proteins in maize clustered into six clades (Fig 4). Interestingly, ZmTIFY1 and 5 clustered with ZmJAZ8, 9, 13, 17, 23, 26, 32 in clade II (Fig 4) while ZmTIFY2 and 3 clustered with ZmJAZ 4, 5, 7, 14, 15, 16, 19, 25 and 37 in clade IV (Fig 4) and the four ZmZML proteins clustered into separate clades (I, II, II, and VI) (Fig 4). Comparison of maize JAZ proteins with orthologues from *Arabidopsis* and sorghum found seven clear groups formed with *Arabidopsis* JAZ proteins clustered into four groups: G1, G3, G4, G5 and G7 while JAZ proteins in maize and sorghum clustered into 6 groups: G2 to G7 (S1 Fig).

## All maize *JAZ* promoters contain JA responsive regulatory elements

Promoter sequences of the *ZmJAZ* genes were analyzed via the PlantCARE database to identify *cis*-regulatory elements in the1.5kb promoter segment upstream of their start codons. Attention was given to elements relevant to hormone and stresses responses (Fig 5; S2 Table). Those elements included: (1) the ABRE motif, involved in abscisic acid (ABA) responses; (2) MBS, a MYB transcriptional factor binding site involved in drought tolerance; (3) MYC, a transcriptional factor of JA responsive genes; (4) the CGTCA- and TGACG-motifs, involved in methyl-JA acid (MeJA) responses; (5) the AuxRR-core, TGA-element, and AuxRE, the auxin-responsive elements; and, (6) the GARE-motif and TATC-box, that serve as gibberellin (GA) -responsive elements.

All putative *ZmJAZ* promoters contained at least two different regulatory elements (Fig 5) with some, such as *ZmJAZ9*, possessing all six elements of the analysis. All *ZmJAZ* promoters contained either MYC-binding site or CGTCA/TGACG–motif for JA and MeJA responses, respetively (Fig 5). Most *ZmJAZ* promoters contained one to several ABRE motifs are involved in abscisic acid (ABA) responsiveness (Fig 5).

## Most *ZmJAZ* genes have non-specific expression across maize tissues under basal conditions

To gain insight into tissue-specific expression of the maize TIFY genes, publicly available transcriptomes of 79 different maize tissues and organs [46] were mined. Of the 43 genes identified in our study, 34 *TIFY* genes were found transcribed at basal levels in at least one of the available tissue types (Fig 6) and several patterns emerged.

Generally, most TIFY genes were found to be expressed in most tissue types, albeit at varying levels of expression. Eight (*ZmJAZ2*, *3*, *8*, *11*, *12*, *13*, *26* and *27*) were found highly expressed (transcript abundance >60 FPKM) and five (*ZmJAZ17*, *25*, *ZmZML1*, *2*, *3*) were

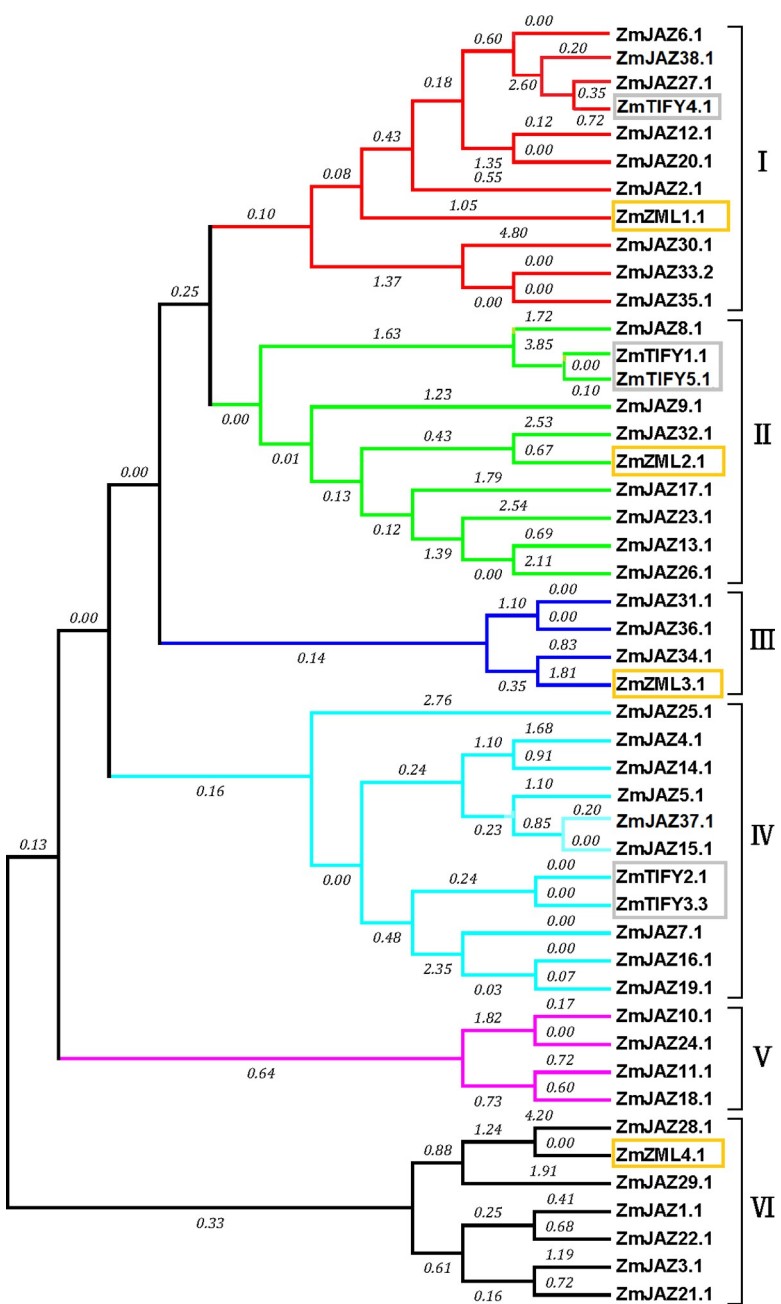

**Fig 4. Phylogenetic analysis of maize JAZ, TIFY and ZML subfamily proteins.** The phylogenetic tree was constructed in software Mega7 by the maximum likelihood method with the bootstrap test of 1000 replicates. Amino acid sequences were aligned with Muscle (Multiple Sequence Comparison by Log-Expectation).

medium-expressed (1<transcript abundance<60 FPKM) across all tissues. (Figs 6 and S2). Seven *TIFY* genes (*ZmJAZ4, 5, 6, 14, 15, 18,* and *21*) were high-expressed but in a limited number of tissue types. Interestingly, tissue specificity was found for three *JAZ* genes; *ZmJAZ14* was found to be seed-specific and *ZmJAZ19* and *22* were only found expressed in the anthers (Figs 6 and S2). Expression of *ZmJAZ4* and JAZ7 was prominent in seed and anther, respectively, but they also displayed low expression across other tissues. *ZmJAZ7, 9, 10, 16, 24, 32, 35,*

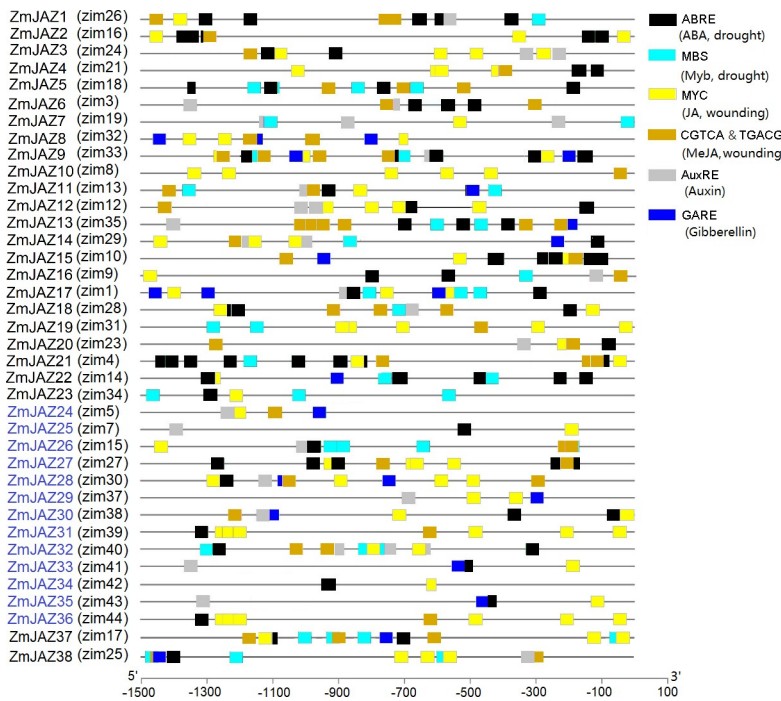

**Fig 5. *cis*-regulatory elements identified in the 1.5 kb-promoter regions of *ZmJAZ* genes.** Different colored boxes denote different type of *cis*-regulatory elements and labeled according to legend. The scale bar indicates the location of each *cis*-elements within the promoters.

*37*, *38*, *ZmTIFY1*, *3*, *4* and *5* were low-expressed genes (transcript abundance < 1 FPKM) in all maize tissues tested under these conditions.

## Expression patterns of *ZmJAZ* genes in response to wounding and JA treatment

To understand the inducibility of *ZmJAZs* during defense responses, we measured transcript accumulation of the gene expression in the second leaves of maize seedlings following mechanical wounding or chemical application. Expression of 18 *ZmJAZ* genes (*ZmJAZ3*, *5*, *6*, *8*, *9*, *11*, *12*, *13*, *15*, *17*, *18*, *20*, *23*, *25*, *31*, *32*, *33*, and *36*) were detected in these experiments.

Apart from *ZmJAZ32 and 36*, all *ZmJAZ* genes tested were found to be transiently induced by wounding. *ZmJAZ5*, *6*, *15*, and *17* displayed a rapid, but short, increase in wound-inducible expression; induction of these genes was detected at 1 and 3 h post-wounding, but subsided by 6 h. Wounding induced expression of *ZmJAZ3*, *8*, *9*, *12*, *18*, *25*, *31*, and *33* as early at 1 h following treatment, but their induction persisted for the duration of the time-course. *ZmJAZ13*, *20*, and *23* appear to be late wound-induced genes and it is likely their peak of their expression was not captured within the time-points tested (Fig 7A).

Mechanical damage induces production of JA and subsequent JA-responsive gene expression. To test the contribution of JA in wound-inducibility of *ZmJAZ* genes, maize seedlings were chemically treated with JA. With the exception of *ZmJAZ15*, *17*, and *20*, most maize *JAZ* genes were found to be JA-inducible. Expression of *ZmJAZ8*, *11*, *12*, *18*, *25*, *31*, *32*, and *33* were induced as early as 6 h and persisted for at least 24 h after treatment. Transcription of *ZmJAZ5*, *6*, and *9* was detected at only 24 hours post wounding. *ZmJAZ15* and *ZmJAZ20* were constitutively expressed in the leaves and unresponsive to JA treatment and remarkably,

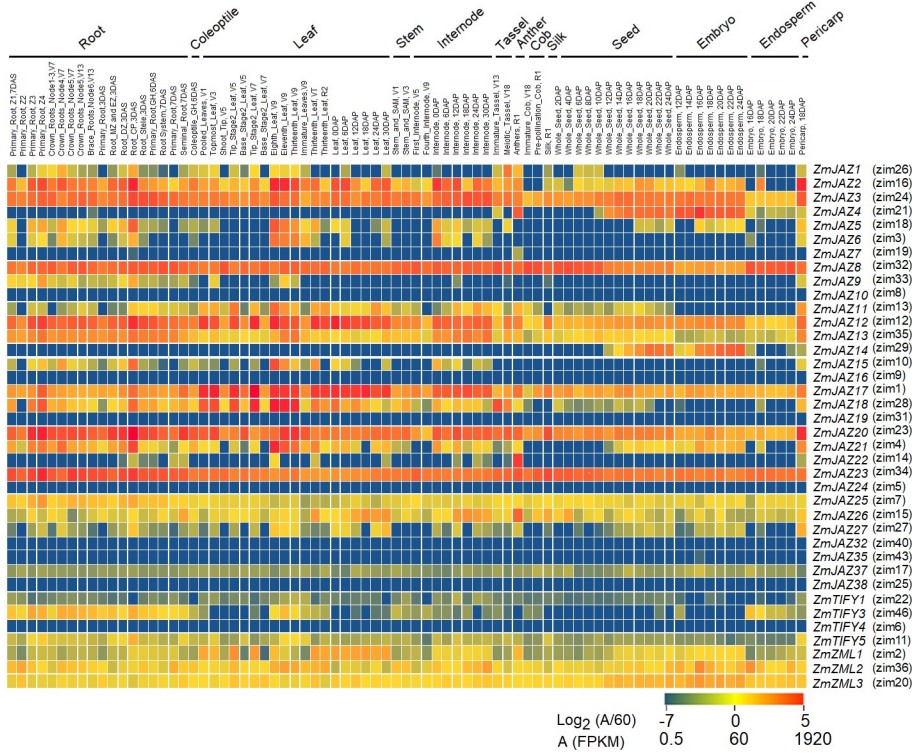

**Fig 6. Heatmap of expression patterns for maize *TIFY* genes in 79 tissues.** The heatmap was generated using log2 (abundance/60) in which red color indicates high expression and blue indicates low expression.

*ZmJAZ17*, the most highly expressed gene in the leaves (Fig 6), was the only *ZmJAZ* observed to be repressed by JA treatment (Fig 7B).

Comparison of expression patterns between mechanically wounded and JA treated plants found that the majority of *ZmJAZ* genes induced by mechanical damage were also JA-responsive. Interestingly, several genes responded differently to JA treatment and wounding in the time-points measured. Wounding, but not JA-treatment, induced *ZmJAZ15* and *20*, however the opposite was observed for *ZmJAZ32* (Fig 7).

## Discussion

In recent years, advances in sequencing technologies have enabled substantial improvements to the maize reference genome providing a more accurate representation of the genomic composition and subsequent gene models. In this study, we used updated B73 reference genome AGPv4 to identify and categorize 47 TIFY family genes, named *ZmZIM1* to *ZmZIM47* (Table 1). This work augments the existing literature with over 40% more maize TIFY members, compared with previous studies that identified only up to 30 isoforms [15,40–42]. More specifically, our analyisis uncovered five, four, and 38 genes to the TIFY, ZML, and JAZ subfamilies, respectively and were named accordingly. No PDD subfamily members were identified during this process, consistent with what is currently understood from other monocot species [15,41]. Compared with other grasses, the maize genome encodes more than twice the number of predicted TIFY genes than *Brachypodium* [36], rice [32], or sorghum [15], and thus far only wheat is known to contain more with 47 identified.

Maize arose from a hybridization of two ancestral species that produced an allotetraploid approximately 14 million years ago and soon after underwent diploidization [52] resulting in a

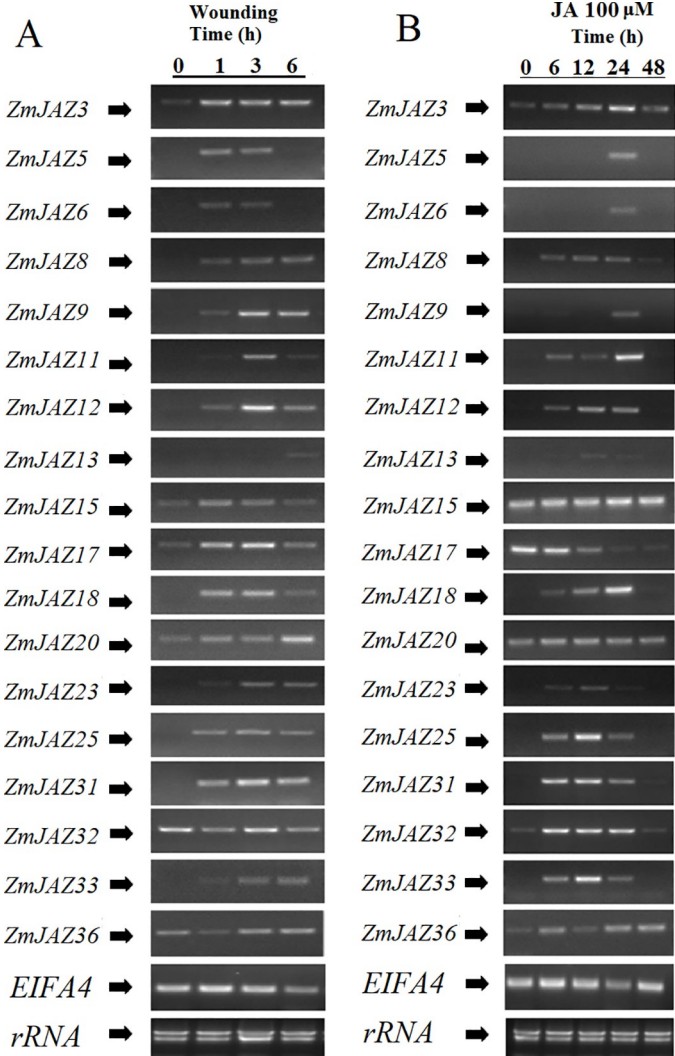

**Fig 7.** Transcription activation of *ZmJAZ* genes upon wounding treatment (A) or 100μM JA treatment (B). Semi-qPCR was conducted to quantify the expression level of the JAZ genes. The *EIF4A* gene was employed as the reference gene.

segmental alleotetraploid [53]. The closest crop species relative to maize is sorghum (*Sorghum bicolor*) which diverged from one of the maize ancestors around the same time of the maize hybridization event. Using the sorghum genome as a guide, segments from the two maize sub-genomes (maize1 and maize2) can be differentiated where each subgenome is orthologous to the sorghum genome [49]. In our analysis, 32 and 14 TIFY genes were identified on the maize1 and maize2 genomes, respectively (Fig 1). This observation agrees with the finding that in modern maize inbred lines, maize2 has exhibited significantly more gene loss compared to maize1 [49].

Prior to the discovery of JAZ proteins [10,12,33], the functional annotation of the plant-specific TIFY family was unclear [13]. [15] analyzed the origin and evolution of the *TIFY* genes and organized them into four subfamilies: ZML, TIFY, PPD and JAZ where the latter can account for 60–80% of the *TIFY* genes in a species and undeniably are the best understood. In *Arabidopsis*, JAZ proteins are transcriptional repressors for JA-mediated response [10–12].

During JA signaling, JA-Ile serves as a ligand to promote the formation of a SCF$^{COI1}$-JA-Ile-JAZ complex in which JAZ proteins are ubiquitinated and subsequently degraded by the 26S proteasome [10,11].

JAZ proteins contain a TIFY and Jas domain in their N- and C-terminus, respectively, and both domains are required during JA signal transduction. The TIFY domain is necessary for homo- and heteromeric dimerization between the TIFY family members [50] and for JAZ-NIN-JA-TPL interaction and the Jas domain is required for the formation of the COI1–JAZ co-receptor complex [51]. In maize, five TIFY subfamily proteins (ZmTIFY1, 2, 3, 4, and 5) contain solely a TIFY domain (S1 Table; Fig 3), however they are highly similar with typical ZmJAZ proteins and cluster with them during phylogenetic analysis (Fig 4). These observations suggest that the maize TIFY subfamily are comprised of JAZ proteins that have lost their Jas domains during the evolutionary process. In contrast, several ZmJAZ proteins possessed normal Jas domains but either lacked (ZmJAZ 29, 30, 31, 32, and 36) or have incomplete (ZmJAZ25 and ZML4) TIFY domains (S1 Table; Fig 3), which likely results in the inability to dimerize normally with other JAZ proteins and the subsequent loss of function as transcriptional repressors.

In terms of the Jas domain, most ZmJAZ proteins were predicted to contain intact sequences (S1 Table), however several isoforms either had an incomplete (ZmJAZ4, 10, and 14) domain lacking the motif "XXPY" or had an altered degron motif (ZmJAZ13, 17, and 26) where the typical "LPIAR" motif was replaced by "VPQAR". In *Arabidopsis*, five JAZ genes encode different transcript variants [12] with *AtJAZ10* having up to four, correspondingly to the protein isoforms, AtJAZ10.1, 10.2, 10.3, and 10.4 [17]. AtJAZ10.3 has a truncated Jas domain missing the "XXPY" motif and AtJAZ10.4 has completely lost Jas domain [12,17]. Loss of normal Jas domain is associated with increased stability during JA signaling process and overexpression of these isoform variants perturb normal JA responses [12,17]. In maize, 14 *JAZ* genes (*ZmJAZ3*, *8*, *11*, *12*, *13*, *17*, *18*, *20*, *23*, *25*, *26*, *27*, *30*, and *33*) have two to seven alternative transcripts, and with several variants missing either TIFY or Jas domain or both (S1 Table). Notably, *ZmJAZ23* showed parallels with *AtJAZ10* in that it encodes several transcript variants, some which produce typical JAZ proteins (ZmJAZ23.1, 23.2, 23.3, and 23.4) and others are either missing (ZmJAZ23.5) have incomplete (ZmJAZ23.6 and ZmJAZ23.7) Jas domains (S1 Table). In summary, this study identified maize JAZ proteins (ZmJAZ4.1, 10.1, 14.1, 23.5, 23.6, 23.7, 27.1, and 33.1) that have Jas domain perturbations likely rendering them resistant to degradation [24] and would have considerable implications for JA desensitization and in physiological processes.

Differential gene expression of large gene families allows plants to finely control their responses with spatial-temporal specificity. In this study, we found that the maize *TIFY* family genes are expressed in a tissue- and organ-specific manner under basal conditions. Eight *JAZ* genes (*ZmJAZ2*, *3*, *8*, *11*, *12*, *13*, *26*, and *27*) were found highly expressed across almost 79 different tissue types while nine others (*ZmJAZ7*, *9*, *10*, *16*, *24*, *32*, *35*, *37*, and *38*) only accumulated transcripts to low levels (Fig 6). Other *ZmJAZ* showed greater tissue specificity: in leaves, 14 *JAZ* genes (*ZmJAZ2*, *3*, *5*, *6*, *8*, *11*, *12*, *13*, *15*, *17*, *18*, *20*, *21*, and *23*) were highly expressed (Fig 6), suggesting that JAZ proteins regulate leaf development and defense. *Arabidopsis* possesses 13 *JAZ* genes [28], 10 of which are essential for vegetative growth and reproductive [29].

Insect herbivory or mechanical damage rapidly induce expression of *JAZ* genes in *Arabidopsis*, and functional analysis with *aos* and *coi1* mutant lines showed that both JA biosynthesis and perception are required in this process [12,30]. In this study, we found that 14 ZmJAZ genes (*ZmJAZ3*, *5*, *6*, *8*, *9*, *11*, *12*, *13*, *18*, *20*, *23*, *25*, *31*, and *33*) are induced by either mechanical wound, exogenous JA application, or both treatments (Fig 7). These results provide pharmacological evidence that JAZ genes from diverse plant species respond by similar cues resulting in similar defensive functions in both monocots and dicots.

Promoter analysis provides insights into the regulation of genes to elucidate their physiological functions. Here, we examined the *ZmJAZ* promoters for six *cis*-regulatory elements involved in defense and hormone responses. ABA facilitates stomatal closure in response to abiotic stress such as during drought conditions [54], while auxin signaling regulates tolerance to diverse stresses [55]. SA, JA, and ET are best understood for their roles in plant defense to diverse biotic and abiotic stresses. Here, they activate transcriptional reprogramming to engage defense against various pathogens, pests, and abiotic stresses, such as wounding and salt [56]. JA and ET usually synergistically regulated plant development and tolerance to necrotrophic fungi [57]. GA is a major growth hormone and stress-induced growth reduction is associated with decreases in GA levels [58]. Our result revealed that *ZmJAZ* gene promoters contain several *cis*-regulatory elements related to plant hormone and stresses regulation. This is in agreement with a recent study that identified *cis*-elements associated with ABA, Auxin, MeJA, GA, and stress tolerances in promoters of wheat *JAZ* genes [43] and consistent with an increasing number of studies that have functionally characterize specific JAZ proteins in plant hormone regulation of defense responses against abiotic and biotic stresses in rice, tomato, maize, and poplar [32,33,42,59]. Thus, it is reasonable to expect that the maize JAZ proteins will emerge as potent mediates in crosstalk hormone signaling crosstalk during plant growth, development, or defense processes.

## Supporting information

**S1 Fig. Phylogenetic tree of JAZ proteins from Maize, *Arabidopsis*, and Sorghum.** The tree was constructed by software Mega7, using the maximum likelihood method with a bootstrap test of 1000 replicates and all the amino acid sequences of JAZ proteins of the three species were aligned with online software Muscle.
(PNG)

**S2 Fig. The expression level of 47 *ZmTIFY* genes in 79 tissues in FPKM value.** The expression data was downloaded from www.maizegdb.org.
(TIF)

**S1 Table. Basic information of the *TIFY* family genes in maize.**
(XLS)

**S2 Table. *cis*-regulatory elements detected within the promoter regions of *ZmJAZ* genes.**
(XLS)

**S3 Table. List of primers used for semi-quantitative PCR.**
(XLS)

**S1 Document. The nucleotide and sequences of TIFY genes in maize at B73 RefGen_v3 andAGPv4.**
(PDF)

**S1 File.**
(ZIP)

## Author Contributions

**Conceptualization:** Yuanxin Yan.

**Formal analysis:** Yannan Shi, Aga Guido Okwana Valerio.

**Funding acquisition:** Yuanxin Yan.

**Investigation:** Pingdong Sun, Yannan Shi, Aga Guido Okwana Valerio, Qingyun Luo, Kang Liu.

**Methodology:** Pingdong Sun, Yannan Shi, Aga Guido Okwana Valerio, Qingyun Luo, Kang Liu.

**Project administration:** Jia Qin.

**Software:** Kang Liu.

**Supervision:** Jia Qin, Yuanxin Yan.

**Visualization:** Qingyun Luo, Kang Liu.

**Writing – original draft:** Aga Guido Okwana Valerio, Yuanxin Yan.

**Writing – review & editing:** Aga Guido Okwana Valerio, Eli James Borrego, Yuanxin Yan.

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
