## [Decision Letter · Decision Letter 0]

24 Nov 2020

PONE-D-20-28741

An updated census of the maize TIFY family

PLOS ONE

Dear Dr. Yan,

Thank you for submitting your manuscript to PLOS ONE. After careful consideration, we feel that it has merit but does not fully meet PLOS ONE’s publication criteria as it currently stands. Therefore, we invite you to submit a revised version of the manuscript that addresses the points raised during the review process.

Please revise the manuscript according to the well-know expert in maize community.

We look forward to receiving your revised manuscript.

Kind regards,

Heping Cao, PhD

Academic Editor

PLOS ONE

Journal Requirements:

2. Please include your tables as part of your main manuscript and remove the individual files. Please note that supplementary tables (should remain/ be uploaded) as separate "supporting information" files

3.PLOS ONE now requires that authors provide the original uncropped and unadjusted images underlying all blot or gel results reported in a submission’s figures or Supporting Information files. This policy and the journal’s other requirements for blot/gel reporting and figure preparation are described in detail at https://journals.plos.org/plosone/s/figures#loc-blot-and-gel-reporting-requirements and https://journals.plos.org/plosone/s/figures#loc-preparing-figures-from-image-files. When you submit your revised manuscript, please ensure that your figures adhere fully to these guidelines and provide the original underlying images for all blot or gel data reported in your submission. See the following link for instructions on providing the original image data: https://journals.plos.org/plosone/s/figures#loc-original-images-for-blots-and-gels.

Reviewers' comments:

Reviewer's Responses to Questions

**Comments to the Author**

1. Is the manuscript technically sound, and do the data support the conclusions?

Reviewer #1: Yes

2. Has the statistical analysis been performed appropriately and rigorously? 

Reviewer #1: N/A

3. Have the authors made all data underlying the findings in their manuscript fully available?

Reviewer #1: Yes

4. Is the manuscript presented in an intelligible fashion and written in standard English?

Reviewer #1: Yes

5. Review Comments to the Author

Reviewer #1: Review of Manuscript Number: PONE-D-20-28741

An updated census of the maize TIFY family

The authors characterize a family of transcription factor genes and build upon previous studies by Bai et al. 2011, Zhang et al. 2015, and Zhou et al. 2015. This study includes the identification of new genes in this family, includes mapping analyses, analyzes splicing variants, examines conserved domains, examines gene expression, and enhances phylogenic analyses.

This work is well documented and presents important novel information that would be of interest to the readers of PLOS One.

There is one major issue with the classification of this gene family in that is does not build upon the earlier work of the Grassius project (Yilmaz et al. 2009. Plant Physiol 149:171-180; https://doi.org/10.1104/pp.108.128579). This project had identified 36 genes in this “super” family (see: https://grassius.org/family.php?family=ZIM&species=Maize) and used the symbols zim1 - zim36 (ZIM = zinc-finger inflorescence meristem) to describe them. While the authors state that the symbol “ZIM” had more recently been replaced by the symbol “TIFY”, this nomenclature developed in 2009 has precedence, represents the official symbols for these 36 genes and is how they are listed at MaizeGDB.org.

Of these 36 genes, all but 4 overlap with the 43 genes predicted in this manuscript.................

See uploaded complete review

6. PLOS authors have the option to publish the peer review history of their article (what does this mean?). If published, this will include your full peer review and any attached files.

Reviewer #1: No

---

## [Author Response · Author response to Decision Letter 0]

1 Jan 2021

Dear editors,

This manucript has been revised according to the reviewer's comments.

Thank you very much for your help.

Yuanxin Yan PhD

---

## [Decision Letter · Decision Letter 1]

18 Jan 2021

PONE-D-20-28741R1

An updated census of the maize TIFY family

PLOS ONE

Dear Dr. Yan,

Thank you for submitting your manuscript to PLOS ONE. After careful consideration, we feel that it has merit but does not fully meet PLOS ONE’s publication criteria as it currently stands. Therefore, we invite you to submit a revised version of the manuscript that addresses the points raised during the review process.

Please make a minor revision as suggested by the expert.

We look forward to receiving your revised manuscript.

Kind regards,

Heping Cao, PhD

Academic Editor

PLOS ONE

Reviewers' comments:

Reviewer's Responses to Questions

**Comments to the Author**

1. If the authors have adequately addressed your comments raised in a previous round of review and you feel that this manuscript is now acceptable for publication, you may indicate that here to bypass the “Comments to the Author” section, enter your conflict of interest statement in the “Confidential to Editor” section, and submit your "Accept" recommendation.

Reviewer #1: All comments have been addressed

2. Is the manuscript technically sound, and do the data support the conclusions?

Reviewer #1: Yes

3. Has the statistical analysis been performed appropriately and rigorously? 

Reviewer #1: N/A

4. Have the authors made all data underlying the findings in their manuscript fully available?

Reviewer #1: Yes

5. Is the manuscript presented in an intelligible fashion and written in standard English?

Reviewer #1: Yes

6. Review Comments to the Author

Reviewer #1: All of the comments I made for the previous version have been addressed. However, in the caption of Table 1: "Five TIFY genes have no typical TIFY domain but include Jas motif." This was not modified. Should this now be modified in light of the four additional genes being presented?

Other than this, I do not see any additional issues. This work is well documented and presents important novel information that would be of interest to the readers of PLOS One. The manuscript should be accepted.

7. PLOS authors have the option to publish the peer review history of their article (what does this mean?). If published, this will include your full peer review and any attached files.

Reviewer #1: No

---

## [Author Response · Author response to Decision Letter 1]

25 Jan 2021

Based on the information the reviewer provided this time, the revised manuscript contained one error in the caption of Table 1. This is an inadvertent mistake of our last revision process. Now, it is corrected. 

We thank a lot for the comments from reviewers and editor.

---

## [Editor Report · Decision Letter 2]

4 Feb 2021

An updated census of the maize TIFY family

PONE-D-20-28741R2

Dear Dr. Yan,

We’re pleased to inform you that your manuscript has been judged scientifically suitable for publication and will be formally accepted for publication once it meets all outstanding technical requirements.

Kind regards,

Heping Cao, PhD

Academic Editor

PLOS ONE
---

## [Editor Report · Acceptance letter]

11 Feb 2021

PONE-D-20-28741R2 

An updated census of the maize *TIFY* family 

Dear Dr. Yan:

I'm pleased to inform you that your manuscript has been deemed suitable for publication in PLOS ONE. Congratulations! Your manuscript is now with our production department. 

Kind regards, 

on behalf of

Dr. Heping Cao 

Academic Editor

PLOS ONE